# Nucleophilic Substitution at Tetracoordinate Sulfur. Kinetics and Mechanism of the Chloride-Chloride Exchange Reaction in Arenesulfonyl Chlorides: Counterintuitive Acceleration of Substitution at Sulfonyl Sulfur by *ortho*-Alkyl Groups and Its Origin

**DOI:** 10.3390/molecules25061428

**Published:** 2020-03-20

**Authors:** Marian Mikołajczyk, Monika Gajl, Jarosław Błaszczyk, Marek Cypryk, Bartłomiej Gostyński

**Affiliations:** 1Department of Organic Chemistry, Centre of Molecular and Macromolecular Studies, Polish Academy of Sciences, 90-363 Lodz, 112 Sienkiewicza, Poland; mgajl@cbmm.lodz.pl (M.G.); blaszcz8@cbmm.lodz.pl (J.B.); 2Department of Structural Chemistry, Centre of Molecular and Macromolecular Studies, Polish Academy of Sciences, 90-363 Lodz, 112 Sienkiewicza, Poland; bgostyns@cbmm.lodz.pl

**Keywords:** sulfonyl chlorides, isotopic chloride exchange, density functional calculations, nucleophilic substitution, reaction mechanisms

## Abstract

The chloride-chloride exchange reaction in arenesulfonyl chlorides was investigated experimentally and theoretically by density functional theory (DFT) calculations. The second order rate constants and activation parameters of this identity reaction were determined for 22 variously substituted arenesulfonyl chlorides using radio-labeled Et_4_N^36^Cl. The chloride exchange rates of 11 sulfonyl chlorides bearing *para*-and *meta*-substituents (σ constants from −0.66 to +0.43) in the aromatic ring followed the Hammett equation with a ρ-value of +2.02. The mono- and di-*ortho*-alkyl substituted sulfonyl chlorides exhibit an enhanced reactivity although both inductive and steric effects lower the reaction rate. The DFT calculations of their structures together with X-ray data showed that an increased reactivity is mainly due to a peculiar, rigid, strongly compressed and sterically congested structure. The DFT studies of the title reaction revealed that it proceeds via a single transition state according to the S_N_2 mechanism. The analogous fluoride exchange reaction occurs according to the addition–elimination mechanism (A–E) and formation of a difluorosulfurandioxide intermediate. The reliability of the calculations performed was supported by the fact that the calculated relative rate constants and activation parameters correlate well with the experimental kinetic data.

## 1. Introduction

The mechanism and steric course of the bimolecular nucleophilic substitution reactions at the sulfur atom (S_N_-S) and their relationship are more complicated than those at the sp^3^-carbon atom [1,2]. As sulfur may form hypervalent tetra- and penta-coordinate compounds (commonly named sulfuranes), [3] the most important problem to solve is whether these S_N_-S reactions occur synchronously according to the classical Ingold’s S_N_2 mechanism via a single transition state (TS) or stepwise by an addition–elimination mechanism (A–E) involving formation of unstable sulfurane as the reaction intermediate (trigonal bipyramidal intermediate, TBPI). They are formed by addition of a nucleophilic reagent (N) to the electropositive sulfur atom in reaction substrates and decompose in the second step by departure of the leaving group (L) to give substitution products (SP).

In an analogy with the S_N_2 substitution at carbon, the S_N_-S reactions at the stereogenic tetrahedral sulfur atom occur also with inversion of configuration at sulfur [4]. However, the stereochemical course of the S_N_-S reactions proceeding according to the addition–elimination mechanism (A–E) depends on the disposal of the entering group (N) and the leaving group (L) in a trigonal bipyramidal structure of a short living sulfur intermediate (TBPI). It is now generally accepted that the diaxial or diequatorial disposal of N and L in a transient sulfurane should lead to inversion of configuration at sulfur, whereas the steric outcome of the axial-equatorial substitution is predicted to be retention (see Scheme 1). Although such a stereochemical course of substitution can be rationalized according to the Westheimer’s concept and his rules (apical entry of N, apical departure of L, microscopic reversibility and pseudorotation of TBPI-C) [5,6], unstable sulfurane intermediates formed in this process have not been detected experimentally [7,8]. Moreover, when substitution at sulfur occurs with inversion of configuration, an unequivocal distinction between a concerted S_N_2 displacement and a stepwise A–E mechanism is difficult.

The major part of extensive investigations on nucleophilic substitution at tricoordinate sulfur compounds was devoted to the stereochemistry-mechanism relationship since these compounds are intrinsically chiral and are known in enantiomeric forms. Our recent results on the acid-catalyzed alcoholysis of sulfinamides and unusual stereochemistry of this reaction are the best illustration of these studies [9]. On the contrary, since the majority of tetracoordinate sulfur compounds are achiral, as for example sulfonyl derivatives, the parallel studies on substitution at the S^IV^-center were mainly concentrated on the kinetics of these reactions and mechanistic implications of the measurements performed. However, in two cases of substitution at the S^IV^-center, in menthyl (*R*)-^16^O,^18^O-phenylmethanesulfonate and (−)-(*R*)-sulfonimidoyl chloride (Scheme 2), the stereochemistry has been established to be inversion [10,11].

The results of very extensive kinetic studies on the nucleophilic substitution reactions at the sulfonyl center and mechanistic conclusions drawn from them have been summarized in an excellent review paper entitled "Sulfonyl Transfer Reactions" published in 1989 [12]. This direction of kinetic investigations is continued until recently with a various intensity [13,14]. Generally, the substitution reactions at the sulfonyl sulfur are discussed in term of the concerted S_N_2-type displacement involving a single transition state which may be tighter or looser depending on the substituents present. The experimental kinetic data for the S_N_2 mechanism were provided by Williams and co-workers [15] who observed a linear Brønsted-type relationship in the transesterification of *p*-nitrophenyl sulfonates with a series of oxyanions. With regard to the sulfonyl transfer reactions, which may occur according to the addition–elimination mechanism (A–E), Page and co-workers [16] provided kinetic data and other arguments which are indicative that some β-sultams (cyclic four-membered sulfonamides) undergo alkaline hydrolysis via a stepwise mechanism involving transient formation of a trigonal bipyramidal sulfurandioxide anion intermediate (TBPI^-^-A). Interestingly, a stable, structurally related sulfurandioxide anion (D) has been synthesized and characterized (X-ray, NMR) by Perkins and Martin (Scheme 3) [17]. The latter is the first example of an isolable intermediate postulated to explain an associative nucleophilic attack at the sulfur atom.

In this place, it is necessary to mention that the claims in the early literature [18,19,20,21,22,23,24] that sulfamoyl and sulfonyl chlorides undergo solvolysis by the dissociative S_N_1-type process were shown to be wrong and have been corrected [25,26,27]. However, in the course of these early studies Vizgert [22,23,24] found that 2,4,6-trimethylbenzenesulfonyl chloride, **(2,4,6-Me_3_)–1** (Scheme 4), as well as other *ortho*-alkyl substituted sulfonyl chlorides, (*R*)-**1** [28], are hydrolyzed faster than benzenesulfonyl chloride **1**. At the same time Bunnett and Bassett [29] came to the conclusion that the two *ortho*-methyls provide no detectable steric hindrance of nucleophilic substitution at the sulfonate sulfur. On the contrary, the ethanolysis of arenesulfinates representing a typical nucleophilic substitution at the sulfinyl center (tricoordinate sulfur) is retarded markedly by the two *ortho*-methyl groups in aromatic ring [30]. 

With the aim to confirm and rationalize the unexpected and surprising reactivity enhancement of the *ortho*-alkyl substituted arenesulfonyl chlorides **1** observed in solvolytic processes, we decided to investigate at first the isotopic chloride-chloride exchange reaction in arenesulfonyl chlorides **1** (Equation (1)) as the simplest model reaction of nucleophilic substitution at the sulfonyl sulfur. This reaction has at least two advantages in comparison with solvolytic processes. First, the exchange reaction can be carried out in aprotic solvents, such as acetonitrile, in which the kinetics is relatively simple because no side or catalytic reactions occur. Secondly, as a result of chemical identity of substrates and products, the relative role of bond making and bond breaking in the transitions state (or intermediate) is eliminated.



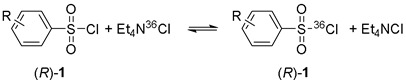
(1)


Moreover, to gain a comprehensive understanding of the reactivity of arenesulfonyl chlorides **1** in the above reaction and to solve a question of its mechanism (S_N_2-S or A–E) the computational studies of this identity reaction were carried out. In this context, it is necessary to mention that Lee and co-workers [31,32] have reported on the DFT studies of the chloride-chloride exchange reaction in methanesulfonyl chloride. This work encouraged us to extend our experimental studies with theoretical calculations especially since in the case of the reaction (1) we are able to confront the results of calculations with experimental findings.

## 2. Results and Discussion

### 2.1. Kinetics of Isotopic Chloride-Chloride Exchange Reaction in Arenesulfonyl Chlorides and Their Reactivity

The standard kinetic measurements of the isotopic chloride exchange between arenesulfonyl chlorides **1** and radio-labeled tetraethylammonium chloride, Et_4_N^36^Cl (Equation 1) were carried out in acetonitrile solution. The progress of the reaction was followed by determining the radioactivity of the water phase containing Et_4_N^36^Cl which was obtained after interruption (freezing) of the reaction and separation of the two reactants by rapid extraction with a water:carbon tetrachloride:methanol mixture in 1:5:0.4 ratio.

In a preliminary investigation [33], the kinetic order and the reactivity of the chloride-chloride exchange of the parent benzenesulfonyl chloride **1** and its three alkyl-substituted analogues, **(4-Me)-1**, **(2,4,6-Me_3_)-1** and **(2,4,6-iPr_3_)-1** were determined. The second order rate constants found for these sulfonyl chlorides are 10 × k = 1.33, 0.67, 6.10 and 3.72 for **1**, **(4-Me)-1**, **(2,4,6-Me_3_)-1** and **(2,4,6-iPr_3_)-1**, respectively. These preliminary data revealed that the presence of the *ortho*-alkyl substituents in the aromatic ring accelerates the exchange process confirming earlier observations on the enhanced reactivity of *ortho*-alkyl substituted sulfonyl chlorides **1** in solvolytic processes.

In the present paper, we report the results of our extended kinetic studies on the reaction under discussion using a large number of substituted arenesulfonyl chlorides **1** (22 examples). They may be divided into two groups. The first one comprised the aromatic sulfonyl chlorides **1** with electron-attracting and electron-donating substituents mainly in the *para* position of the aromatic ring. In the second group there were *ortho*-alkyl substituted sulfonyl chlorides **1**. The latter were investigated to determine/explain their unexpected high reactivity displayed in our identity reaction.

The observed rates (k_ex_) of the exchange reaction (Equation 1) were always of the first order with respect to each of the reactants (sulfonyl chloride **1** and ^36^Cl^-^ ion). The second order rate constants at 0 °C (k_0_) for all the investigated chlorides **1** together with the Arrhenius activation parameters (log A and E) are summarized in Table 1. In Table 2 the rate constants at 25 °C (k_25_) as well as the Eyring activation parameters are presented. The results collected in Table 1 and Table 2 allowed us to determine the kinetic effect of substituents in the aromatic ring of **1** on the exchange reaction. The σ constants of the selected eleven substituents were in a wide range from −0.660 for 4-Me_2_N to 0.430 for 3-CF_3_ group. The Hammett plots (Figure 1 and Figure 2) were linear at both investigated temperatures. The rate constants k_25_ were found to correlate better (r = 0.993) with σ substituents constants than the rate constants k_0_ (r = 0.973). The calculated reaction constants are ρ_0_ = 1.94 ± 0.24 (Figure 1) and ρ_25_ = 2.02 ± 0.07 (Figure 2), respectively. 

The effect of substituents on rate of the chloride exchange reaction is apparent from Table 1 and Table 2. Thus, electron-attracting substituents increase the rate of the investigated reaction while electron-donating substituents slow it down. For example, the chloride exchange occurs 12 times slower in **(4-Me_2_N)-1** than that in benzenesulfonyl chloride **1**. On the other hand, **(3-CF_3_)-1** is ca. 10 times more reactive in the chloride exchange reaction in comparison with unsubstituted **1**. At first sight, the observed strong accelerating or retarding effect of substituents may be best rationalized in terms of a concerted one-step substitution at sulfur of the S_N_2-type with a symmetrical transition state. Moreover, the positive Hammett rho-value (ρ = 2.02) is in accord with the S_N_2 mechanism and indicates that the reaction center is very sensitive to the electronic effect of substituents [34]. However, the relatively large rho value (ρ > 2) found in our reaction may also be indicative of a stepwise addition–elimination mechanism (A–E) as was proposed by the Italian group [35].

As these two mechanisms cannot be distinguished unequivocally by the kinetics, especially in the case of our very simple identity reaction, we decided to solve this problem by combination of the experimental kinetic data presented herein with the extensive DFT studies of the chloride exchange reaction in arenesulfonyl chlorides.

In the second set of kinetic measurements the effect of alkyl substituents in the benzene ring of **1** on the rate of the isotopic chloride-chloride exchange was determined. It was hoped that the results obtained (Table 1 and Table 2) would help, at least partially, in elucidation of the astonishing increased reactivity of the *ortho*-alkyl substituted sulfonyl chlorides **1**. In general, the alkyl substituents in the aromatic ring may exert two kinds of effect on the reactive sulfonyl center, namely inductive effect and steric effect. Both of them are expected to decrease the reactivity of sulfonyl chlorides **1** and the rate of the chloride exchange. The reactivity order of mono-alkyl substituted **1**, with the exception of *ortho*-substituted analogues is shown below.

**1****(3-Me)-1****(4-Me)-1****(3,4-Me_2_)-1**10^2^ × k_25_117.3102.067.064.4

The values of k_25_ shown above demonstrate clearly that the inductive effect of the methyl substituents lowers the exchange rate, as expected. In this context, more interesting and also more informative is the reactivity of 2-alkyl and 2,4,6-trialkyl substituted benzenesulfonyl chlorides **1** (see below).

**(2-Et)-1****(2-Me)-1****(2-iPr)-1****1**
10^2^ × k_25_248.3128.1124.0117.3

**(2,4,6-Me_3_)-1****(2,4,6-Et_3_)-1****(2,4,6-iPr_3_)-1****1**10^2^ × k_25_517.8446.6378.4117.3

All three 2-alkyl substituted sulfonyl chlorides **1** were found to be more reactive than the unsubstituted one. The most reactive in the chloride exchange reaction are 2,4,6-trialkyl substituted **1**. They are 3 to 5 times more reactive than unsubstituted **1**. Inspection of their reactivities indicates that the classical steric effect (steric hindrance) reduces k_25_ after switching from methyl to isopropyl. Thus, in spite of the fact that both inductive and steric effects act together in one direction to lower the exchange rate, the results briefly discussed above point to a noticeably accelerating effect of the *ortho*-alkyl substituents.

In accord with this view are the results with **2-CD_3_**- and **4-CD_3_**-substituted benzenesulfonyl chlorides **1** which are also presented in Table 1 and Table 2. It was found that accelerating effect of the *ortho*-CD_3_ group is slightly stronger than that of the *ortho*-CH_3_ group as indicated by the higher rate constant k_25_ = 157.6 for **(2-CD_3_)-1** compared to k_25_ = 128.1 for **(2-CH_3_)-1**. This difference in the rate constants may be best explained on the basis of the fact that the effective size of the CD_3_ group is smaller than that of the CH_3_ group [36]. Recently, secondary kinetic isotope effect was also investigated in solvolysis and alkoholysis of mesitylene sulfonyl chloride **(2,4,6-Me_3_)-1** and its deuterated analogue **(2,4,6-(CD_3_)_3_)-1 [14]**. However, comparable rate constants were observed for both deuterated and undeuterated chlorides **1**. In our opinion, the solvation-desolvation effects of polar protic solvents are much stronger in solvolytic reactions of sulfonyl chlorides and most probably cover the small differences in reactivity resulting from SKIE.

Elucidation of enhanced reactivity of *ortho*-substituted benzenesulfonyl chlorides is important not only from the scientific point of view. For a long time 2,4,6-triisopropylbenzenesulfonyl chloride, **(2,4,6-*i*Pr_3_)-1**, and its trimethyl analogue, **(2,4,6-Me_3_)-1**, have found wide use as the best coupling reagents in oligo- and polyribonucleotide synthesis [37,38,39]. They have been selected experimentally, however, the question why they are the best is still open.

The reasons for the observed acceleration are till now not fully understood. In our preliminary communication [33] we expressed the view that this effect may be attributed to relief of steric interactions between alkyl groups and the sulfonyl oxygen atoms upon transformation of the tetrahedral sulfonyl chloride structure into a trigonal bipyramidal intermediate. Later on, the increased reactivity of *ortho*-alkyl substituted benzenesulfonyl chlorides, defined as “steric acceleration”, was claimed to be due to relief of strain in the transition state having trigonal bipyramidal structure [40]. More recently, it was proposed that the structure of reactants and molecularity of the transition state with nucleophilic assistance of solvent may be responsible for the increased reactivity of hindered sulfonyl chlorides [13].

### 2.2. Structural Features of Arenesulfonyl Chlorides 1: DFT Calculations and X-Ray Diffraction Data

To gain a deeper understanding of the effect of *ortho*-alkyl groups on the kinetics of substitution at the sulfonyl sulfur, we carried out a DFT study of the identity chloride exchange reaction. As the reactivity of molecules is controlled by their electronic and three-dimensional structures, and, as it was demonstrated above, the steric hindrance of *ortho*-alkyl groups lowers the reactivity of sulfonyl chlorides **1** in our reaction, we first examined the calculated structures of arenesulfonyl chlorides and compared them to the known crystal structures, whenever possible.

In this part of our study several arenesulfonyl chlorides were investigated: unsubstituted **1**, **(4-Me)-1**, **(2,6-Me_2_)-1**, **(2,4,6-Me_3_)-1**, **(2,6-iPr_2_)-1** and **(2,4,6-iPr_3_)-1**. The latter chloride, being the most computationally demanding, was calculated only with the simpler method (at the double-zeta level, see Computational Methods). The more accurate calculations were performed for 2,6-di(isopropyl) analogue which was expected to show a similar reactivity level. In addition, calculations were also carried out for C_6_F_5_SO_2_Cl (**F_5_-1**) in order to examine the influence of the exceptionally strong electron-withdrawing substituent on the reaction energetics.

The geometries of model benzenesulfonyl chlorides as well as of the stationary points on the substitution pathway were calculated by three density functional methods: (i) B3LYP with Grimme’s empirical correction for dispersion forces (denoted as B3LYP-GD3) with double-zeta 6-31+G(d) basis set; (ii) B3LYP-GD3 with triple-zeta 6-311+G(2d,p) basis set; and (iii) APFD functional with dispersion treatment included and 6-311+G(2d,p) basis set (see Computational Methods section). It was stated previously that the correct modeling of sulfur compounds requires f polarization functions included in basis set [41]. Our tests with 6-311++G(3df,2p) basis set revealed that the results were qualitatively the same and the improvement in geometries is not significant. Therefore, it was decided to use the smaller 6-311+G(2d,p) basis set in order to save computing time. The accuracy of applied theoretical methods in reproducing geometries of sulfonyl chlorides was verified by comparing the optimized geometries with experimentally determined crystal structures. Thus, the following reported X-ray structures were used as references: 4-methylbenzenesulfonyl chloride (**(4-Me)-1**) [42], 2,4,6-trimethylbenzenesulfonyl chloride (**(2,4,6-Me_3_)-1**) [14] and 2,4,6-tri(isopropyl)benzenesulfonyl chloride (**(2,4,6-iPr_3_)-1**) [43,44]. Comparison of the calculated and experimental geometrical parameters is presented in Figure 3 and Figure 4 and in Appendix A.

The calculated geometries are in good agreement with the X-ray structures ((Figure 3, Figure 4 and Figure 5), Appendix A). Comparison reveals that (i) all calculation methods generally provide slightly too long bond distances; (ii) triple-zeta basis set provide more accurate bonding parameters than the double-zeta set; (iii) APFD gives somewhat better agreement of geometrical parameters than B3LYP-GD3. However, due to numerical problems with optimization of some transition structures with APFD, we chose B3LYP-GD3/6-311+G(2d,p) for use throughout the text, with the exception of **(2,4,6-iPr_3_)-1**, which was optimized with the B3LYP-GD3/6-31+G(d) method only. The calculated C-S and S-Cl bonds in **(2,6-Me_2_)-1**, **(2,4,6-Me_3_)-1**, **(2,6-iPr_2_)-1** and **(2,4,6-iPr_3_)-1** are longer than in unsubstituted **1**, because of steric requirements of *ortho* alkyls, while the S-O bonds as well as the C-C bonds in the ring practically do not change. The inductive and resonance effect of *para*-methyl group on the bonding in sulfonyl chlorides manifests itself by a slight shortening of the C-S and S-O bonds compared to the corresponding benzenesulfonyl chlorides having no methyl group in *para* position, i.e., **(4-Me)-1** vs. **1** and **(2,4,6-Me_3_)-1** vs. **(2,6-Me_2_)-1** (Appendix A). In acetonitrile solution, the polar S-Cl bonds are slightly longer than in the gas phase especially in transition states and in reactant complexes, due to interaction with polar solvent. Bond angles are almost unaffected by the solvent.

A similar spatial situation is also observed in symmetrical trimethyl analogue, (**(2,4,6-Me_3_)-1**).

Due to the presence of the two *ortho*-isopropyl groups in **(2,4,6-iPr_3_)-1** the aromatic ring is no longer planar and adopts a strongly flattened boat conformation. Moreover, the sulfonyl group is sterically fixed by two weak hydrogen bonds formed between the sulfonyl oxygen atoms and both methine hydrogens of the isopropyl groups (benzylic hydrogens). Therefore, it cannot freely rotate around the C-S bond. The formation of weak hydrogen bonds between sulfonyl oxygens and secondary hydrogens of the isopropyl groups has been supported by geometrical parameters and by Natural Bond Orbital (NBO) analysis. The distances between H atoms in the *ortho*-alkyl groups and oxygen atoms of the sulfonyl moiety are 2.15–2.3 Å and the C-H-O angles 119–122° (SM, Appendix A). An example of the orbital interaction n_O_→σ*_H-C_ as calculated by the NBO method is shown in Appendix A.

The other consequence of the presence of *ortho*-alkyl substituents is a twist of the SO_2_Cl fragment relative to the ring plane. While benzenesulfonyl chloride **1** shows C_s_ symmetry with the S-Cl bond lying in the plane perpendicular to the ring plane (to be precise: the plane defined by Cl, S and C1 atom of the ring is perpendicular to the ring plane), in *ortho*-substituted sulfonyl chlorides **(2,6-Me_3_)-1**, **(2,4,6-Me_3_)-1**, **(2,6-iPr_2_)-1**, **(2,4,6-iPr_3_)-1** and **(F_5_)-1** the SO_2_Cl group is slightly twisted off the perpendicularity. A slight deviation from perpendicularity is also observed in the crystal structure of **(4-Me)-1** which is probably induced by the interactions (packing forces) in the crystal lattice. The deviations from perpendicularity measured as the dihedral angle between S-Cl bond and the ring plane is in the range of 0–14 ° depending on the size of *ortho* substituents and to some extent on the calculation method and on the quality of basis set (Appendix A). Based on the X-ray data the following dihedral angles were calculated: 14°, 5° and 5.5° for **(2,4,6-iPr_3_)-1**, **(2,4,6-Me_3_)-1** and **(4-Me)-1**, respectively. These values are also in agreement with those obtained from DFT calculations. However, it should be noted that this effect is not significant energetically.

Dissecting the results of our calculations and X-ray data of mono-and di-*ortho*-alkyl substituted arenesulfonyl chlorides **1** and taking into account the results of our reactivity studies presented above several important observations and interpretations can be inferred. First of all, the increased level of reactivity of di-*ortho*-alkyl substituted arenesulfonyl chlorides **1** in nucleophilic substitution reactions can be attributed to a particular structure which they adopt both in a solid state and in solution. Due to the hydrogen bond formation discussed above, these sulfonyl chlorides **1** exist in one rigid most reactive conformation in which there is no rotation around the C(1)-S bond. The *ortho*-alkyl substituents involved in the hydrogen bond formation with the SO_2_Cl group are also sterically fixed. Moreover, the hydrogen bond formation results in a substantial compression around the sulfonyl moiety and steric congestion. Therefore, in our opinion, the release of internal strains present in the tetrahedral structure of *ortho*-alkyl substituted **1** on going to the trigonal bipyramidal transition state or sulfurane intermediate is the major factor responsible for the increased reactivity of sulfonyl chlorides **1** bearing one or two alkyl groups in *ortho*-position.

Finally, the efficacy of 2,4,6-triisopropylbenzenesulfonyl chloride **(2,4,6-iPr_3_)-1**, (TrisCl—commercial abbreviation), as activating agent in the synthesis of oligonucleotides deserves a concise comment. As it was demonstrated herein by us and earlier also by others, this sulfonyl chloride, like other di-*ortho*-alkyl substituted analogues, exhibits an enhanced reactivity in a nucleophilic substitution at the sulfonyl sulfur. This property of TrisCl is very important for the synthesis of nucleotides because in the formation of the internucleotide bond two nucleophilic substitution reactions at the sulfonyl sulfur take place. In the first step, TrisCl reacts with strongly nucleophilic heterocyclic amines (2-methylimidazole or 3-nitro-1H-1,2,4-triazole) and pyridine as an additional nucleophilic catalyst. Then, the corresponding sulfonamides or N-sulfonylpyridinium salt formed, which are more reactive than TrisCl, give upon treatment with the properly protected nucleoside phosphate salt the corresponding mixed phosphoric and sulfonic acid anhydride as a result of the second displacement reaction at sulfur. In the last synthetic step, the mixed anhydride reacts at phosphorus with the free hydroxyl group of a protected nucleoside molecule to give protected dinucleoside phosphate. All the three reactions were found to occur very quickly affording the desired product in a virtually quantitative yield [39]. In this context, it is interesting to point out that the present knowledge of the factors affecting the reactivity of arenesulfonyl chlorides allows to design new more reactive coupling reagents. Thus, for example, one can expect that 2,6-dimethyl-4-trifluoromethylbenzenesulfonyl chloride, **(2,6-Me_2_-4-CF_3_)-1**, will be more reactive than trimethyl analogue **(2,4,6-Me_3_)-1** because of the presence of the electron-attracting para-CF_3_ group which significantly accelerates nucleophilic substitution at the sulfonyl sulfur.

### 2.3. DFT Study of Identity Chloride Exchange Reaction in Arenesulfonyl Chlorides 1

As it was mentioned earlier, the gas-phase halide exchange reactions (Cl,Cl), (F,F) in methanesulfonyl halides were theoretically investigated by Lee et al. [32]. For the identity chloride exchange reaction the DFT calculations showed a double-well potential energy surface (PES) with a central transition state surrounded by the reactant and product ion-dipole complexes. Such a shape of PES corresponds to a typical S_N_2 gas-phase energy profile. On the other hand, the fluoride exchange reaction was found to proceed via a triple well PES (two transition states) with the difluorosulfurane intermediate formed according to the addition–elimination mechanism (A–E). In this way, the authors demonstrated a possibility of distinguishing between the synchronous substitution (S_N_2) at the sulfonyl sulfur and a stepwise A–E mechanism. As the results of kinetic studies described above were not conclusive with regard to the mechanism of our identity reaction (Eq.1), we decided to solve this problem using DFT theoretical calculations.

Our calculations revealed that the chloride exchange reaction in benzenesulfonyl chloride **1** proceeds analogously to MeSO_2_Cl. The incoming Cl^-^ forms at first an ion-dipole complex at a rather long distance from S (r(Cl-S) = 3.075 Å in **1**) which upon further approach is transformed into a transition state with both r(S-Cl) distances equal. In the gas phase the energy of TS is lower than the sum of the energies of individual substrates due to the energy gain resulting from the stabilization of the negative charge in the complex compared to the isolated chloride anion. In solution, the negatively charged nucleophile is stabilized by a polar solvent. The substitution results in elevation of the TS energy relative to the substrates, because of a partial desolvation of reagents in the transition state. Nucleophilic substitution in MeCN solution shows only the transition state without distinct minima. Careful inspection of a PES scan in solution indicates, however, that the corresponding ion-dipole complexes are also formed, which are a few kcal/mol higher in energy than the sum of substrates and are connected with them via a very small energy barrier of only ca. 0.2 kcal/mol. The distance between the sulfur atom and approaching Cl^-^ in the reactant complexes (RC) in solution is significantly longer than in the corresponding RC complexes in the gas phase [4.097 Å (MeCN) vs. 3.075 Å (gas) in **1···Cl^−^ (RC)**] due to the effect of polar solvent which induces the increased separation of ion from the dipole fragment. Structures of PhSO_2_Cl (**1**), PhSO_2_Cl_2_^-^, reactant complex (**1···Cl^−^ (RC)**) and transition state (**[1-Cl]^−^ (TS)**) (C_2v_ symmetry) optimized by B3LYP-GD3/6-311+G(2d,p) method in gas phase are shown in Figure 6. Comparison of selected bonding parameters in arenesulfonyl chlorides **1**, complexes and transition states is presented in Table 3. The calculated structures of transitions states are illustrated in Figure 7. The two apical chlorines and central S atoms in the transition states are not collinear, i.e., the bond angle <Cl-S-Cl ranges from ca. 167.5° to 168.8° (163.5° in **F_5_-1**), because of repulsive effects between the two apical negatively charged Cl atoms and the basal oxygen atoms and of the n_Cl_→π*_C-C_ delocalization, according to NBO second order perturbation theory analysis (Appendix A).

The Cl-S-Cl axis in the transition state **[1-Cl]^−^ (TS)** is perpendicular to the plane defined by the aromatic ring. The same symmetry is found for the Cl exchange in **(4-Me)-1**. In all *ortho*-alkyl substituted arenesulfonyl chlorides, the Cl-S-Cl axis is deviated from the perpendicular direction. The corresponding transition state structures are shown in Figure 7.

Surprisingly, in the case of the transition state for the reaction of pentafluorophenyl sulfonyl chloride, **F_5_-1**, with Cl^-^ the Cl-S-Cl axis is perpendicular to the ring plane, both in the gas phase and in solution, despite of the expected electrostatic repulsion between sulfonyl oxygens and fluorines in the ring.

Free energy profiles for the Cl exchange in PhSO_2_Cl in the gas phase and in MeCN solution are compared in Figure 8 and the enthalpies of complex formation in gas phase and Gibbs free energy barriers ΔG^‡^ for the Cl exchange reaction in MeCN solution are collected in Table 4.

It is very important to note that the relative rate constants calculated with two methods for the chloride identity exchange reaction in solution correlate well with the experimental data. This supports the reliability of calculations and indicates that the kinetic reactivity can be rationalized on the basis of theoretical calculations.

For comparison with the chloride exchange reaction, analogous exchange of fluoride in benzenesulfonyl fluoride **2** was calculated. In contrast to chloride, the fluoride exchange in sulfonyl fluorides proceeds according to the addition–elimination mechanism as it was reported for MeSO_2_F [32]. The addition–elimination pathway of the fluoride exchange is in line with the generally accepted view that strongly electron-withdrawing substituents are beneficial to a stepwise reaction through an intermediate [3]. In the gas phase a symmetrical difluorosulfurane intermediate (SI) is being created in which distances of both fluorine atoms from sulfur are identical. For **2** the complex has C_2v_ symmetry, the same as the transition state in chloride exchange in **1**. For *ortho*-substituted sulfonyl fluorides, the F-S-F axis is no longer perpendicular to the ring plane. For example, for **[(2,4,6-Me_3_)-2-F]^-^** (SI) the deviation from perpendicularity is ca. 18°. In MeCN solution a sulfurane intermediate exists as well but with higher energy than the reactants, which indicates a minor energy barrier that is associated with a change in system solvation. We failed to find the structures of these transition states because of the very flat potential energy surface. The reaction profiles ΔG_rel_ (kcal/mol) for fluoride exchange **2** in the gas phase and in solution are shown in Figure 9.

An analogous behavior is characteristic of pentafluorobenzenesulfonyl chloride (**F_5_-1**) and fluoride (**F_5_-2**). The chloride exchange in **F_5_-1** occurs via a transition state with the free energy barrier ΔG^‡^ = 1.6 kcal/mol with respect to the reactant complex (in the gas phase) and ΔG^‡^ = 16.2 kcal/mol with respect to the reactants in MeCN solution. The fluoride exchange occurs via the difluorosulfurane intermediate with the free energy ΔG = −38.5 kcal/mol lower than the reactants in the gas phase and ΔG = 0.05 kcal/mol higher than the reactants in MeCN. In the intermediate (SI) for fluoride exchange, both for the gas phase and for solution, the O-S-O plane deviates by 13.5° (gas phase) and 21° (MeCN) from the plane of the ring (Figure 10).

To explain the apparent order of reactivities, the steric and electronic factors associated with substitution in the benzene ring were analyzed by Natural Bond Orbital (NBO) theory and by isodesmic reactions.

Second order perturbation analysis within the frame of Natural Bond Orbital theory is a convenient method to examine the electron delocalization pattern in the molecule [45,46]. NBO analysis predicts that the electronic structure of the sulfonyl group in methanesulfonyl derivatives is highly polarized with significant contributions from reciprocal hyperconjugation of the form n→σ*, where substituents around the central sulfur act as donors and acceptors at the same time. In contrast, the d-orbital participation in these systems seems to be insignificant [41]. The importance of hyperconjugation in organic sulfonyl chlorides was later confirmed by sulfur K-edge X-ray absorption spectroscopy (XAS) [47].

We applied the NBO method to methyl-substituted benzenesulfonyl chlorides to see how rotation of sulfonyl group relative to phenyl ring plane affects the electron delocalization in the molecule. The results are summarized in Appendix A. Total electron delocalization between benzene ring and sulfonyl chloride moiety (SO_2_Cl↔ring) was measured by deletion of all interactions between two corresponding atom blocks [45]. In all compounds under study it is relatively weak. The strongest delocalizations (besides π_CC_→π*_CC_ aromatic delocalizations in the ring) are those in which oxygen lone pairs act as electron donors, i.e., n_O_→σ*_SO_, n_O_→σ*_SCl_ and also n_O_→d_S_. Lone pairs of Cl participate as donors in hyper-conjugation to much smaller extent than the oxygen lone pairs. Surprisingly, in contrast to the previous report [41], a relatively high contribution of the n_O_→d_S_ delocalizations has been observed.

Basic observations may be summarized as follows:Rotation of sulfonyl group around C(1)-S bond within the observed range does not noticeably affect the energies of electron delocalization in terms of the NBO theory (Appendix A, Appendix A).Variation in steric repulsion between SO_2_Cl moiety and aromatic ring in **1** upon its rotation around C(1)-S bond is also insignificant.The NBO analysis does not provide any clear rationalization for the observed kinetic effect. The differences in delocalization pattern between simple benzenesulfonyl chlorides **1** and **(4-Me)-1** and sterically hindered **(2,6-Me_2_)-1, (2,4,6-Me_3_)-1, (2,6-*i*Pr_2_)-1** are not significant.

In order to estimate the energy of steric congestion induced by *ortho*-alkyl groups in sulfonyl chlorides and its change during the exchange reaction, we calculated the energies of isodesmic reactions involving substituted and unsubstituted sulfonyl chlorides with chloride anion (Scheme 5). The reaction (5a) allows estimation of the change in the strain energy (and, to some extent, in the electronic effects) upon conversion of 4-MePhSO_2_Cl, **(4-Me)-1**, into 2,6-Me_2_PhSO_2_Cl, **(2,6-Me_2_)-1**. The reaction (5b) allows calculation of the strain energy as the energy of analogous conversion proceeding between the corresponding transition states. The reactions (5c) and (5d) provide the estimates of the inductive effect of the methyl group in *para* position relative to sulfonyl group compared to unsubstituted PhSO_2_Cl (**1**) and the same effect in the transition state of the chloride exchange. Reactions (5e) and (5f) allow estimation of the steric energy of the two *ortho*-isopropyl groups in **(2,6-iPr_2_)-1** and in the transition state of the chloride exchange reaction.

Table 5 collects the calculated enthalpies and Gibbs free energies for the reactions 5a–f in Scheme 5. Comparison of the free energies of reactions 5a and 5b shows that the energy of reaction 5b involving redistribution of Me groups in the transition states is lower than free energy of reaction 5a involving parent sulfonyl chlorides (ΔG = 3.4 kcal/mol in 5b vs. 8.1 kcal/mol in 5a in the gas phase and 7.6 vs. 9.7 kcal/mol in solution). This result can be interpreted as a partial release of internal strain upon transformation of tetrahedral sulfonyl chloride into a trigonal bipyramid of the transition state, [**(2,6-Me_2_)-1-Cl]**^−^**-TS**, in accord with our early suggestion [33] and a later proposal [40]. This effect should result in elevated reactivity of *ortho*-alkyl substituted sulfonyl chlorides. Reactions 5c and 5d illustrate the electronic (inductive) influence of *para* methyl group to molecular energy. The Gibbs free energy of this exchange in sulfonyl chloride (Equation 4) is 0.4 kcal/mol which would suggest a weak unfavorable effect. In the transition state the corresponding free energy is 1.8 kcal/mol (Equation (5)) indicating that the *para*-methyl group slightly increases the free energy barrier compared with the parent chloride **1** by ca. 1.4 kcal/mol in the gas phase. This should result in lower reactivity of *para*-methylbenzenesulfonyl chloride, **(4-Me)-1**, compared to **1**. One can also notice that the steric energy of the two isopropyl groups in *ortho* positions in **(2,6-*i*Pr_2_)-1** is associated with a greater steric energy than the two methyl groups by ca. 1.8 kcal/mol. The release of steric energy in the transition state under gas phase conditions is only 2 kcal/mol compared to 4.7 kcal/mol for methyl groups. Nevertheless, in this case the strain release is also predicted to reduce the free energy barrier and to enhance the reactivity. Interestingly, the strain release in the transition state is not related to the O^…^HC hydrogen bond breaking. Both the O-H distance (ca. 2.20 Å) and weak n_O_→σ*_CH_ delocalization interactions are very similar as in parent sulfonyl chlorides.

The results of the above analysis are consistent with the conclusions obtained from structural and kinetic considerations presented above.

### 2.4. Concluding Remarks

The present study was undertaken with two major aims. The first was to rationalize an unexpected high reactivity of mono- and especially di-*ortho*-substituted arenesulfonyl chlorides observed in various nucleophilic substitution reactions at the sulfonyl sulfur atom. The second task was to find a new way for distinguishing between two mechanisms (S_N_2 or A–E) of bimolecular nucleophilic substitution at the sulfur atom. To solve both problems the identity chloride exchange reaction in arenesulfonyl chlorides as a simple model reaction was investigated both experimentally (reaction kinetics) and theoretically (DFT calculations).

The kinetic measurements showed that steric effects of alkyl substituents in the aromatic ring, including also *ortho*-alkyl substituted chlorides, lower the reactivity. Our DFT calculations of the ground-state structures of selected arenesulfonyl chlorides and their literature X-ray data clearly indicated that the increased level of reactivity of di-*ortho*-substituted sulfonyl chlorides is due to a unique structure they adopt in a solution and solid state. These sulfonyl chlorides exist in one rigid most reactive conformation held by two intramolecular hydrogen bonds formed between the sulfonyl oxygen atoms and the α-C-H hydrogens of the *ortho*-alkyl groups. In such a structure, there are substantial internal strains connected with a compression around the sulfonyl moiety and steric congestion. The release of the above internal strains on going to the trigonal bipyramidal transition state (or sulfurane intermediate) is the major factor responsible for the increased reactivity of sulfonyl chlorides under discussion. Therefore, the words “steric acceleration” and “positive steric effect” to describe the *ortho*-alkyl group effect in substitution at tetracoordinate sulfur are not proper and misleading. It is worthy of note that knowledge of the factors affecting the reactivity of arenesulfonyl chlorides is important for their future rational applications in other fields.

The second question connected with the mechanisms (S_N_2 or A–E) of the nucleophilic substitution at sulfur in our identity chloride exchange reaction was solved by a combination of the DFT calculations with the results of kinetic studies reported herein. Our calculations showed that for the chloride exchange reaction in benzenesulfonyl chloride and other substituted analogues a double-well PES with a single transition state surrounded by reactant and product ion-dipole complexes was observed. Such a shape of PES means that the chloride-chloride exchange proceeds synchronously according to the S_N_2 mechanism. It is important to note that the calculated rate constants and activation parameters for the chloride exchange reaction correlate well with the experimental data. For comparison purposes, similar calculations for the fluoride exchange reaction revealed that it proceeds according to the A–E mechanism with the formation of difluorosulfurandioxide anion as intermediate since a triple-well PES with two transition states was observed.

The present study demonstrates that kinetic investigations coupled with DFT calculations can be used as a convenient method to establish mechanism of nucleophilic substitution reaction not only at sulfur but also at other heteroatoms.

## 3. Materials and Methods

### 3.1. Materials

Arenesulfonyl chlorides were purchased from chemical companies (Fluka AG (Buchs, Switzerland)), KochLight (Gauteng, South Africa), Sigma-Aldrich (St. Louis, MO, USA), Merck (Darmstadt, Germany), POCh-Polish Chemical Reagents (Gliwice, Poland), or prepared according to the literature [48]. They were purified by fractional distillation or crystallization. Their melting or boiling points and n_D_ values were compared with the literature data [48,49]. In many cases [22,23] the purity of sulfonyl chlorides was confirmed by elemental analysis.

Radio-labeled tetraethylammonium chloride was obtained by neutralizing aqueous solution of hydrochloric acid, H^36^Cl, with aqueous solution of tetraethylammonium hydroxide, Et_4_NOH (Fluka AG). Water was then removed and the salt was crystallized several times from ethanol and dried in vacuum over P_2_O_5_. Tetraethylammonium chloride, Et_4_N^36^Cl obtained in this way, is very hygroscopic and was stored over P_2_O_5_.

Acetonitrile (Merck) was refluxed over P_2_O_5_ and fractionally distilled twice through a long Widmer column. The middle fraction boiling at 80 °C was collected.

### 3.2. Kinetic Methods

The isotopic chloride exchange experiments were carried out in acetonitrile solution with reacting mixtures of 0.01 M radioactive ammonium salt and 0.02 M sulfonyl chloride. The temperature was maintained with accuracy of ±0.1 °C. The reaction was initiated in a glass ampoule of about 2.5 mL by a quick addition of a solution (1 mL) of sulfonyl chloride **1** in acetonitrile at the reaction temperature. The reaction was stopped by freezing the sample in liquid nitrogen. Then the reactants were separated by a quick extraction using a mixture of water (1 mL) carbon tetrachloride (10 mL) and methanol (0.4 mL). In an independent experiment, it was checked that none of the sulfonyl chlorides investigated hydrolyzed during 2 minutes extraction and the chloride ions, Cl^-^, passed completely to the water phase. The radioactivity of these samples of water phase (0.5 mL) was determined by the liquid scintillation counter Intertechnique SL30 (France) using the scintillator p-terfenyl (6 g/L) and naphthalene (50 g/L) in 1,4-dioxane solution. The exchange reaction was usually followed till 2–4 half-lives.

### 3.3. Kinetic Calculations

The exchange rate constant, k_ex_, and their statistical errors, Δk_ex_, were calculated by means of linear regression [50,51] on the confidence level 0.95 from the plots of log(*I_t_* − *I_∞_*) against time, where *I_t_* and *I_∞_* are the counting rates of the sample at time *t* and ∞, respectively. The second-order rate constants, k, were then calculated from the equation $k=\frac{k_{ex}}{a+b}$, where *a* and *b* are the concentration of the reagents.

The calculations of the Arrhenius activation parameters log *A* and *E* were carried out by the method of weighted regression using statistical weights $\frac{k^{2}}{{\left(\Delta k\right)}^{2}}$. From the Arrhenius plot the rate constants k_25_ at 25 °C were calculated for all the sulfonyl chlorides. In the same way, the rate constants k_0_ at 0 °C were calculated.

The enthalpy, $\Delta H_{25}^{\neq }$, and entropy, $\Delta S_{25}^{\neq }$, of activation were calculated according to the Eyring theory. Free energy of activation,$\;\Delta G_{25}^{\neq }$, was calculated from the experimental data. All the statistical errors were given at the confidence level 0.95.

### 3.4. Computational Methods

All quantum mechanical calculations were performed using the Gaussian 09 suite of programs [52]. Geometries of the model compounds were optimized using two DFT methods: the hybrid B3LYP density functional [53] corrected for dispersion interactions using Grimme GD3 empirical term [54] and APFD functional [55], with the 6-31+G(d) and 6-311+G(2d,p) basis sets in the gas phase and in acetonitrile solution. All stationary points were identified as stable minima by frequency calculations, both in the gas phase and in solution. The vibrational analysis provided thermal enthalpy and entropy corrections at 298 K within the rigid rotor/harmonic oscillator/ideal gas approximation [52]. Thermochemical corrections were scaled by a factor of 0.98. In the cases when double-zeta basis set was used, the more accurate single point electronic energies were obtained using the B3LYP functional, including Grimme GD3 dispersion correction, with the triple-zeta 6-311+G(2d,p) basis set. This level of theory is denoted as B3LYP-GD3/6-311+G(2d,p)//B3LYP-GD3/6-31+G(d). SCRF calculations in acetonitrile were performed using CPCM model with UFF atomic radii as implemented in Gaussian 09 [52]. Free energies in solution were calculated from vibrational analysis using rigid rotor/harmonic oscillator/ideal gas approximation.

Using the rigid rotor/harmonic oscillator/ideal gas method (RRHOIG) to calculate ΔG in solution is often criticized as inappropriate [56]. However, recent studies suggest that the RRHOIG approximation in many cases provides surprisingly good results [57,58].

Population analysis was performed using the NBO 6.0 module as the external procedure in GAUSSIAN 09 at the RHF/6-311+G(2d,p) level of theory [45,46].

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
