# Peer review of "Nucleophilic Substitution at Tetracoordinate Sulfur. Kinetics and Mechanism of the Chloride-Chloride Exchange Reaction in Arenesulfonyl Chlorides: Counterintuitive Acceleration of Substitution at Sulfonyl Sulfur by ortho-Alkyl Groups and Its Origin"

_molecules, 2020, doi:10.3390/molecules25061428_

Round 1

Reviewer 1 Report

One of the notable achievements of physical organic chemistry is our understanding of nucleophilic substitution reactions at carbon; namely, SN1 and SN2 mechanisms. Attempts to expand such mechanistic understanding to nucleophilic substitution at phosphorus and sulfur has been of great interest. Here there are additional possibilities such as addition generating a pentavalent intermediate followed by elimination to generate the substituted product. For example phosphate derivatives have been extensively studied and show the formation of addition products, which may pseudorotate followed by elimination. Similar studies have been done with sulfonyl chlorides and the results have been reviewed. However, the results are very complicated owing to nucleophile, substrate and solvent  effects and not well understood as outlined in the review papers cited. For example, in 1975 Mikolajczyk et al. reported the surprising result that o-alkyl groups accelerated nucleophilic substitution at sulfur of arylsulfonyl chlorides in acetonitrile in a 2nd order reaction despite the expected deceleration due to steric and electronic effects. The basis for this result has intrigued and mystified researchers for years. However, new and reliable computational methods provide insight into these conundrums. In the present manuscript these results are expanded and extensively investigated computationally using DFT. The studies are well done and lead to the conclusion that the acceleration by o-isopropyl groups is due to "weak hydrogen bonds between the sulfonyl oxygens and secondary hydrogens of the isopropyl groups" which results in compression and steric congestion which is relieved on going to the trigonal bipyramidal transition state. This conclusion leads to the question of the effect of substituting D for H at the secondary hydrogens. This is addressed earlier in the manuscript in which o-CD3 has a greater accelerating effect than o-CH3 but 2,4,6-Me3-and its deuterated analogue have comparable rate constants. This result is ascribed to solvation-desolvation effects. To support this conjecture perhaps using DFT calculations incorporating solvent continuums are needed.

Listed below are some suggested minor corrections:

line 79 methane

line 110 insert "the" before reaction

line 182 delete "it"

line 231 change "Since" to "For"

line 232 delete "a" after "found"

line 334 delete "for" after "deserves"

line 483 change"estimate" to "estimation"

line 485 change "to calculate" to "calculation"

line 489 change "to estimate" to "estimation"

line 502 change "in" to "with"

Author Response

Reviewer's comment:

"...the question of of the effect of substituting D for H at the secondary hydrogens. (...) This result is ascribed to solvation-desolvation effects. To support this conjecture perhaps using DFT calculations incorporating solvent continuums are needed."

Answer

Our calculations were performed in acetonitrile solution modelled within the continuous medium approximation (CPCM). Therefore the solvent effect is implicitly included.
We focused on a group of specific sulfonyl chlorides and deuterated ones were out of scope of this research. However we agree that DFT calculations of solvation affects could help in better understanding these observations.

All minor corrections were made accordingly and highlighted in the corrected manuscript.

Reviewer 2 Report

The article entitled "Nucleophilic Substitution at Tetracoordinate Sulfur. Kinetics and Mechanism of the Chloride-Chloride Exchange Reaction in Arenesulfonyl Chlorides: Counterintuitive Acceleration of Substitution at Sulfonyl Sulfur by ortho-Alkyl Groups and Its Origin" is a very nice investigation concerning the substitution reaction at sulfur. The article is well written and organized. However, I think the introductions is quite long and should be shortened. In addition, I would like that the authors observe the following points:

1) Page 1, line 40: since TBPI is firstly described on this page, I would like to suggest the authors to include the meaning for it: TBPI (Trigonal bipyramidal intermediate).

2) Page 2, line 44: I was wondering if the authors meant "stereochemical course" instead of "steric course" of the SN-S reactions. It seems to me that they are actually describing the stereochemical outcome. Please, check this out.

3) Page 2, line 82: (-)-R-sulfnimidoyl should be (-)-R-sulfnimidoyl

4) Page 5, line 160: 0oC should be corrected to 0 oC. Also, on the same page (line 161) 25oC should be corrected to 25 oC.

5) One last point: On page 10 (line 299) the authors mentioned that "the aromatic ring adopts a strongly flattened boat conformation". The authors should clarify it since the aromatic ring is essentially plane. 

Author Response

Reviewer 2

"the introduction is quite long and should be shortened"

Answer

It has been shortened by ca. half a page. However, the introduction is addressed not only to heteroatom chemists but also to physical organic as well as synthetic chemists who may be not familiar with the problems of nucleophilic substitution at heteroatoms.
All minor corrections were made accordingly and highlighted in the corrected manuscript.
Ad 3) the name "(-)-R-sulfonimidoyl" is correct
Ad 5) we added additional expression to this sentence hoping that now it is more clear.